# IMPROVING ADVERSARIAL TRAINING WITH MARGIN-WEIGHTED PERTURBATION BUDGET

## ABSTRACT

Adversarial Training (AT) effectively improves the robustness of Deep Neural Networks (DNNs) to adversarial attacks. Generally, AT involves training DNN models with adversarial examples obtained within a pre-defined, fixed perturbation bound. Notably, individual natural examples from which these adversarial examples are crafted exhibit varying degrees of intrinsic vulnerabilities, and as such, crafting adversarial examples with fixed perturbation radius for all instances may not sufficiently unleash the potency of AT. Motivated by this observation, we propose a simple, computationally cheap reweighting function for assigning perturbation bounds to adversarial examples used for AT. We name our approach *Margin-Weighted Perturbation Budget (MWPB)*. The proposed method assigns perturbation radii to individual adversarial samples based on the vulnerability of their corresponding individual natural examples. Experimental results show that the proposed method yields a genuine improvement in the robustness of existing AT algorithms against various adversarial attacks.

## 1 INTRODUCTION

In recent years, Deep Neural Networks (DNNs) have demonstrated remarkable success across various domains, achieving impressive performance benchmarks. However, this success has come hand in hand with a critical concern: the vulnerability of DNNs to well-crafted adversarial perturbations Szegedy et al. (2013); Goodfellow et al. (2014). This observed brittleness has raised questions about the safe deployment of DNNs in safety-critical applications.

In response to the challenge of adversarial vulnerability, a multitude of defense mechanisms have been proposed to enhance the robustness of neural networks. Among these, adversarial training (AT) Goodfellow et al. (2014); Madry et al. (2017) stands out as one of the most prominent and effective approaches. AT typically involves training neural networks using adversarial examples. The effectiveness of AT has inspired many variants such as Zhang et al. (2019); Wang et al. (2019); Ding et al. (2019); Zhang et al. (2020); Zeng et al. (2021), among others.

Moreover, alternative methods, such as adversarial weight perturbation Wu et al. (2020), instance reweighting techniques Zhang et al. (2020); Liu et al. (2021); Fakorede et al. (2023b), and hypersphere embedding methods Pang et al. (2020); Fakorede et al. (2023a), have emerged to further enhance the performance of existing AT variants. These diverse strategies collectively aim to fortify the resilience of DNNs against adversarial threats.

It has been established that the efficacy of adversarial training varies significantly across samples of various classes Xu et al. (2021); Fakorede et al. (2023b); Wei et al. (2023). Adversarial examples derived from natural samples that inherently possess are significantly misclassified in adversarially trained models Liu et al. (2021); Zhang et al. (2020). Specifically, the robust accuracy achieved when evaluating adversarial samples originating from natural examples closer to class boundaries is substantially lower than those achieved with adversarial samples stemming from inherently more robust natural examples. In response to these challenges, various reweighting techniques have been introduced to enhance the effectiveness of AT Zhang et al. (2020); Liu et al. (2021); Fakorede et al. (2023b). These techniques operate by assigning greater weights to or prioritizing the loss of disadvantaged examples.

It's important to highlight that current AT methods predominantly rely on adversarial examples generated with predetermined, fixed perturbation radii. However, the notable performance variations observed across different adversarial examples prompt a fundamental question: *Is uniform perturbation of adversarial examples used in AT necessary or beneficial, as is commonly practiced in existing research?* For instance, *should adversarial examples crafted from inherently vulnerable natural samples be allocated the same perturbation budget as those derived from naturally more robust examples?*

In this work, we present a case against applying uniform perturbation. We demonstrate that first-order adversarial attacks, such as projected gradient descent (PGD), induce a higher increase in adversarial loss for adversarial samples originating from vulnerable natural examples compared to those derived from inherently robust natural examples when subjected to uniform perturbation radii. Enlarging the perturbation radii may increase the inner maximization loss of adversarial examples derived from inherently robust natural examples. Consequently, we introduce a reweighting method designed to allocate perturbation budgets to individual adversarial examples employed in adversarial training. Our proposed method assigns these budgets based on the vulnerabilities exhibited by their corresponding natural examples. We employ a logit margins approach to estimate each natural sample's vulnerability.

Our rationale is as follows: Natural examples that are initially misclassified or located in proximity to the decision boundary may require only minimal perturbations to yield effective adversarial examples suitable for training. Conversely, when crafting adversarial examples from intrinsically robust natural examples, larger perturbations may be necessary to achieve better training impact. Experimental results show that the proposed method improves the performance of existing AT methods including standard AT Madry et al. (2017), TRADES Zhang et al. (2019), and MART Wang et al. (2019).

We summarize the contributions of our work as follows:

1. We argue for assigning varying perturbation radii to individual adversarial samples based on the vulnerability of their corresponding natural examples in the inner maximization component of the min-max adversarial training framework. Consequently, we propose a reweighting function for assigning perturbation radii to individual adversarial examples crafted for adversarial training.

2. We empirically demonstrate the effectiveness of the proposed strategy in improving adversarial training.

3. We show the superiority of the proposed method over existing reweighting and adaptive perturbation radii methods, especially against stronger white-box and black-box attacks.

## 2 RELATED WORK

### 2.1 ADVERSARIAL ROBUSTNESS.

Adversarial robustness refers to a model's ability to withstand adversarial attacks. Many methods Guo et al. (2018); Papernot et al. (2017); Madry et al. (2017); Goodfellow et al. (2014); Zhang et al. (2019) have been proposed to improve adversarial robustness of neural networks. Nevertheless, some of these methods are ineffective against stronger attacks. Adversarial training (AT) Madry et al. (2017), which involves training the model with adversarial examples obtained under worst-case loss, has significantly improved robustness. Formally, AT involves solving a min-max optimization as follows:

$$\min_{\boldsymbol{\theta}} \mathbb{E}_{(\mathbf{x},y)\sim\mathcal{D}} \left[ \max_{\mathbf{x}'\in B_\epsilon(\mathbf{x})} L(f_{\boldsymbol{\theta}}(\mathbf{x}'), y) \right] \tag{1}$$

where $y$ is the true label of input feature $\mathbf{x}$, $L()$ represents the loss function, $\boldsymbol{\theta}$ are the model parameters, and $B_\epsilon(\mathbf{x}) : \{\mathbf{x}' \in \mathcal{X} : \|\mathbf{x}' - \mathbf{x}\|_p \le \epsilon\}$ represents the $l_p$ norm ball centered around $\mathbf{x}$ constrained by radius $\epsilon$. In Eq. (1), the inner maximization tries to obtain a worst-case adversarial version of the input $\mathbf{x}$ that increases the loss. The outer minimization then tries to find model parameters that would minimize this worst-case adversarial loss. The relative success of AT has inspired an array of variants including Zhang et al. (2019); Wang et al. (2019); Wu et al. (2020); Pang et al.

(2020), to cite a few. A prominent variant TRADES (Zhang et al., 2019) employs a regularization term that trades off adversarial robustness against accuracy. MART (Wang et al., 2019) utilizes a regularization term that explicitly leverages misclassified examples.

## 2.2 REWEIGHTING

Recent works have advocated for assigning unequal weights to the inner maximization loss Zeng et al. (2021) and robust losses Liu et al. (2021); Zhang et al. (2020) to improve the performance of AT. It has been shown that adversarially-trained models poorly classify adversarial examples crafted from natural examples that are intrinsically harder to classify Xu et al. (2021); Fakorede et al. (2023b). Most existing reweighting methods attempt to improve AT by upweighting the robust losses corresponding to vulnerable adversarial examples. While some of these methods improve model robustness to certain attacks, they have performed quite poorly in defending against strong attacks Fakorede et al. (2023b).

In contrast with existing reweighting approaches focusing on reweighting losses, this paper uniquely proposes reweighting the perturbation radii of adversarial examples used for AT. In addition, unlike previous works that attempt to improve robust accuracy by assigning larger weights to examples closer to the decision boundary, this work aims to improve AT by assigning larger perturbation budgets to adversarial examples crafted from inherently robust natural examples.

## 2.3 ADAPTIVE PERTURBATION RADII.

Few works in literature have been directed at adaptive perturbation radii for adversarial training. Notably, (Ding et al., 2019) propose a method that maximizes margin to achieve robustness while also adaptively selecting the "correct" perturbation radius for each data point. The "correct" radius for each data point is characterized by the "shortest successful perturbation" to misclassify the data point. Unlike (Ding et al., 2019), which searches for the shortest successful perturbation, our approach assigns significantly large perturbation radii to intrinsically robust examples. Similarly, (Balaji et al., 2019) propose a method that begins with using a perturbation size $\epsilon_i$ that is as large as possible, then adaptively adjusts the $\epsilon_i$ depending on whether PGD succeeds in finding a misclassified label at $\epsilon_i$. Unlike these works that perform an exhaustive search for suitable perturbation radii, our proposed method simply computes the logit margin on individual natural examples and utilize this information to reweight perturbation radii at no additional cost. Lastly, these works fail to show that improved adversarial robustness against strong adversarial attacks may be achieved using adaptive perturbation radii for adversarial training.

## 3 NOTATIONS AND PRELIMINARIES

We use bold letters to represent vectors. We denote $\mathcal{D} = \{\mathbf{x}_i, y_i\}_{i=1}^n$ a data set of input feature vectors $\mathbf{x}_i \in \mathcal{X} \subseteq \mathbf{R}^d$ and labels $y_i \in \mathcal{Y}$, where $\mathcal{X}$ and $\mathcal{Y}$ represent a feature space and a label set, respectively.

Let $f_\theta : \mathcal{X} \to R^C$ denote a deep neural network (DNN) classifier with parameters $\theta$, and $C$ represents the number of output classes. For any $\mathbf{x} \in \mathcal{X}$, let the class label predicted by $f_\theta$ be $F_\theta(\mathbf{x}) = \arg\max_k f_\theta(\mathbf{x})_k$, where $f_\theta(\mathbf{x})_k$ denotes the $k$-th component of $f_\theta(\mathbf{x})$. $f_\theta(x)_y$ is the probability of $x$ having label $y$.

We denote $\| \cdot \|_p$ as the $l_p$- norm over $\mathbf{R}^d$, that is, for a vector $\mathbf{x} \in \mathbf{R}^d, \|\mathbf{x}\|_p = (\sum_{i=1}^d |\mathbf{x}_i|^p)^{\frac{1}{p}}$. An $\epsilon$-neighborhood for $\mathbf{x}$ is defined as $B_\epsilon(\mathbf{x}) : \{\mathbf{x}' \in \mathcal{X} : \|\mathbf{x}' - \mathbf{x}\|_p \leq \epsilon\}$. An adversarial example corresponding to a natural input $\mathbf{x}$ is denoted as $\mathbf{x}'$. We often refer to the loss resulting from the adversarial attack (inner maximization) as adversarial loss. Also, we interchangeably refer to a DNN trained using adversarial examples as robust network or adversarially trained network.

## 4 PROPOSED METHOD

In this section, we motivate and present a novel perturbation budget assignment function for the inner maximization of the AT framework.

### 4.1 A Case Against Fixed Perturbation.

AT employs a min-max optimization procedure aimed at minimizing worst-case losses computed over perturbed adversarial examples. While the application of AT significantly enhances model robustness, it is essential to acknowledge that the optimized worst-case losses represent approximations that may sometimes be suboptimal Mao et al. (2023). This limitation arises from the nature of the Projected Gradient Descent (PGD) algorithm, which is prone to converging to multiple local maxima scattered across the optimization landscape Madry et al. (2017).

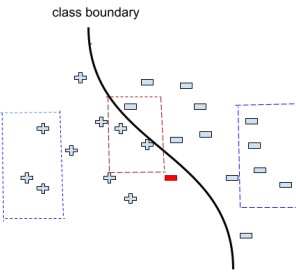

class boundary

Figure 1: Pictorial illustration of robust, vulnerable, and misclassified natural data points. Robust and vulnerable natural data points are in the blue and red dotted boxes respectively. Misclassified points are colored red.

Furthermore, the geometric characteristics inherent to individual natural data samples utilized in the inner maximization step of the min-max optimization exhibit substantial variation. Some natural samples inherently fall into misclassification regions or exhibit elevated loss values. For these particular examples, the PGD algorithm may readily identify suitable maxima. Conversely, naturally robust samples may not yield the worst-case loss under similar optimization settings as their more vulnerable counterparts. A pictorial depiction of the geometric characteristics of natural data samples is given in Figure 1.

The pursuit of adversarial examples that optimally maximize losses represents a compelling objective for achieving highly robust models. Nevertheless, it is important to recognize that the prevailing practice involves applying uniform optimization settings for the inner maximization step, irrespective of the individual idiosyncrasies inherent to each original sample. Specifically, it is commonly assumed that the maximal loss for each example can be discovered using the same perturbation radius. However, it becomes evident that, under this uniform perturbation radius, the losses observed for adversarial examples derived from challenging natural samples exhibit an increased discrepancy compared to those computed for natural samples situated farther from the decision boundary. This assertion warrants a theoretical examination, which we delve into below.

**Theorem 1** *Let $\mathcal{L}$ and $f_\theta(.)$ denote the cross-entropy loss function and the predictions of the model respectively. Consider two natural input-label pairs $(x_1, y_1)$ and $(x_2, y_2)$ that are inherently vulnerable and robust respectively such that $\mathcal{L}(x_1, y_1) > \mathcal{L}(x_2, y_2)$. The following holds for first-order adversarial examples crafted from $x_1$ and $x_2$ within the same perturbation radius $\epsilon$:*

1. $\mathcal{L}(f_\theta(x_1'), y_1) > \mathcal{L}(f_\theta(x_2'), y_2)$

2. $\mathcal{L}(f_\theta(x_1'), y_1) - \mathcal{L}(f_\theta(x_1), y_1) > \mathcal{L}(f_\theta(x_2'), y_2) - \mathcal{L}(f_\theta(x_2), y_2)$

*Remark.* Theorem 1 underscores that when subjected to the same perturbation radius, adversarial examples stemming from vulnerable with high loss values natural examples incur a relatively more substantial loss increase. Similarly, when subjected to a fixed perturbation radius, adversarial examples generated from robust natural examples with low natural loss values induce relatively smaller loss increments than adversarial examples from vulnerable natural examples. Furthermore, it can be inferred that under uniform perturbation, adversarial examples from various natural examples have varying loss increments. We provide proof of Theorem 1 in Appendix B.

Finding adversarial examples with better maxima (higher loss values) is associated with better adversarial robustness Madry et al. (2017). Therefore, using a uniform perturbation radius for the inner-maximization may not yield the best robustness. The reason is due to the considerable variance in the inner maximization losses of individual examples under uniform perturbation radius. Additionally, considering that adversarial examples originating from inherently robust (low-loss) natural examples tend to result in relatively smaller increases in loss, we suggest an approach where we increase the perturbation radii when generating adversarial examples from robust natural examples and decrease the perturbation radii for adversarial examples derived from vulnerable or misclassi-

fied natural examples. In Theorem 2, we demonstrate that enlarging the perturbation radii around an example can indeed lead to an increase in the loss.

## 4.2 Margin-based Reweighting for Perturbation Radii

In the preceding section, 4.1, we presented a rationale against the use of uniform perturbation radii for crafting adversarial examples employed in AT. Instead, we advocate for assigning distinct perturbation radii for generating adversarial examples during the inner maximization step based on the inherent vulnerabilities of their original natural examples. We propose measuring these vulnerabilities in terms of the proximity of the natural data to the class boundaries.

Measuring the exact proximity of a data point to the decision boundary is not straightforward for non-linear models like DNNs. We adopt a measure of *multi-class margin* described in (Koltchinskii & Panchenko, 2002) to estimate the vulnerability or robustness of natural examples. Consider the predictions of a DNN denoted by $f_\theta$ and a labelled example $(x, y)$, the margin $d_{margin}(x, y; \theta)$ is given as follows:

$$d_{margin}(\mathbf{x}, y; \theta) = f_\theta(\mathbf{x})_y - \max_{j, j \neq y} f_\theta(\mathbf{x})_j \qquad (2)$$

where $f_\theta(\mathbf{x})_y$ is the model's predicted probability of the correct label $y$, and $\max_{j, j \neq y} f_\theta(\mathbf{x})_j$ is the largest prediction of the remaining classes.

We utilize the information provided by $d_{margin}(\mathbf{x}, y; \theta)$ in measuring the vulnerability of a natural input example **x** as follows:

- If $d_{margin}(\mathbf{x}, y; \theta) > 0$, **x** is correctly classified and scored. We consider **x** relatively robust.
- If $d_{margin}(\mathbf{x}, y; \theta) = 0$, it implies that **x** has the same prediction score as the best of the remaining classes. As such, we consider **x** to be located at the class boundary.
- If $d_{margin}(\mathbf{x}, y; \theta) < 0$, **x** is considered to be vulnerable, since it is located in a wrong region even before **x** is adversarially perturbed.
- Lastly, in addition to indicating whether a sample **x** is vulnerable or not, we determine degree of vulnerability based on the magnitude of the value returned by $d_{margin}(\mathbf{x}, y; \theta)$. For instance, if $d_{margin}(\mathbf{x}_1, y_1; \theta) > d_{margin}(\mathbf{x}_2, y_2; \theta)$, we consider $\mathbf{x}_2$ to be relatively more vulnerable than $\mathbf{x}_1$.

In Theorem 1, we establish that the inner maximization process inherently induces a more substantial increase in loss for samples that possess intrinsically high loss values. Naturally, this holds true for misclassified natural samples, characterized by negative margins and exhibiting high loss values. Furthermore, considering that these misclassified examples are already situated within regions of high loss, we advocate for a strategy in which the inner maximization process for misclassified examples employs smaller perturbation radii.

Conversely, the inner maximization process inherently leads to a relatively lower increase in loss for samples with low loss values, typically associated with positive margins. Consequently, we propose the utilization of larger perturbation radii for generating adversarial examples from these low-loss samples. As a result, we introduce a radius reweighting function formulated as follows:

$$\epsilon_i = exp(\alpha \cdot d_{margin}(\mathbf{x}_i, y_i; \theta)) * \epsilon \qquad (3)$$

where $\epsilon = 8/255$, the commonly used fixed perturbation radius parameter, and $\alpha$ is a hyperparameter that controls the weight of the function.

The perturbation radius reweighting function in Eq. 3 serves to allocate larger perturbation radii ($\epsilon_i > 8/255$) when generating adversarial examples from natural samples characterized by positive margins, which are correctly labeled. In contrast, it assigns smaller perturbation radii ($\epsilon_i < 8/255$) when crafting adversarial examples from samples with negative margins, which are misclassified natural samples. Additionally, it's important to note that adversarial examples originating from samples situated exactly at the class boundary are crafted using the default perturbation radii ($\epsilon_i = 8/255$).

**Theorem 2** *The inner maximization* $\max_{\mathbf{x}' \in B_\epsilon(\mathbf{x})} L(f_\theta(\mathbf{x}'), y)$ *increases as $\epsilon$ increase.*

We provide the proof in the Appendix B.

The main idea behind *AT* is to minimize the maximum loss incurred by adversarial examples within a specified budget. However, it becomes apparent that the magnitude of loss increase resulting from the inner maximization step varies based on the losses of the original examples. These variations may indicate that adversarial examples generated from inherently robust natural examples may not exhibit a sufficiently high loss for achieving optimal robustness. Conversely, the losses of adversarial examples stemming from misclassified natural examples may be excessively high and exhibit poor generalization, as discussed in prior work Balaji et al. (2019). The larger perturbation radii assigned for generating adversarial examples from robust natural examples through the proposed reweighting function in Eq. 3 plays a crucial role in achieving an adversarial loss that surpasses what can be attained through inner maximization alone under a fixed perturbation radius of $\epsilon = 8/255$.

### 4.3 APPLYING THE MARGIN-WEIGHTED PERTURBATION BUDGET (MWPB) METHOD

Every adversarial training method is a variant of min-max optimization. Hence, our proposed reweighting method may be applied to any adversarial training variant. We re-write the min-max adversarial training objective in Eq. 1 as follows:

$$\min_{\boldsymbol{\theta}} \frac{1}{n} \sum_{i=1}^{n} \left[ \max_{\mathbf{x}'_i \in B_{\epsilon_i}(\mathbf{x}_i)} L(f_\theta(\mathbf{x}'_i), y_i) \right] \tag{4}$$

where each $\epsilon_i$ is computed according to Eq. 3 for each input-label pair $(\mathbf{x}_i, y_i)$, and $B_{\epsilon_i}(\mathbf{x_i})$ : $\{\mathbf{x}'_i \in \mathcal{X} : \|\mathbf{x}_i' - \mathbf{x}_i\|_p \leq \epsilon_i\}$. For the purpose of our experiments, which we present in Section 5, we combine our approach with popular existing AT variants standard *AT* Madry et al. (2017), *TRADES* Zhang et al. (2019), and *MART* Wang et al. (2019). We respectively termed them *MWPB-AT*, *MWPB-TRADES*, and *MWPB-MART*.

### 4.4 CHALLENGE OF ADVERSARIAL TRAINING WITH LARGER PERTURBATION RADII

The adversarial loss landscape is unfavorable to optimization under large perturbation budgets Liu et al. (2020). It is shown that when a perturbation size $\epsilon$ is large, the gradients become small due to decreased gradient magnitude in the initial sub-optimal region, making it challenging for the model to escape the sub-optimal initial region. Large perturbation size may also encourage a model to find sharper minima. In contrast, smaller perturbation budgets facilitate larger gradient magnitude in the initial sub-optimal regions, which in turn help prevent the model from getting stuck in these regions.

Given that our proposed method requires perturbing a subset of the training data with relatively larger perturbation budgets, we use a two-phase training approach. We train the initial epochs with adversarial examples obtained under a smaller perturbation budget of $\epsilon/2$. This allows the model to gradually adapt. Subsequently, we transition to the adversarial training objective outlined in Eq. 4, which employs larger perturbation budgets. The pseudo-code for the proposed algorithm is presented in the Appendix C.

## 5 EXPERIMENTS

In this section, we extensively evaluate the proposed method. To test the versatility of our method, we test on various datasets including CIFAR-10 (Krizhevsky et al., 2009), SVHN (Netzer et al., 2011), and TinyImageNet (Deng et al., 2009). We utilize Resnet-18 He et al. (2016) and Wideresnet-34-10 He et al. (2016) as the backbone models. For brevity, we respectively denote ResNet-18 and Wideresnet-34-10 as RN18 and WRN34-10.

### 5.1 EXPERIMENTAL SETUP

**Training Parameters.** We trained the networks using mini-batch gradient descent for 110 epochs, with momentum 0.9 and batch size 128. We use the weight decay of 5e-4 for training CIFAR-10 and 3.5e-3 for SVHN and Tiny Imagenet. The initial learning rate is set to 0.1 (0.01 for SVHN

and Tiny Imagenet), and divided by 10 in the 80-th epoch, and then at the 90-th epoch. We train the first 80 epochs with adversarial examples obtained via PGD with a smaller perturbation budget of $4/255$ and step size of $1/255$. Subsequently, we introduce MWPB-AT, MWPB-TRADES, and MWPB-MART in the 81-st epoch to improve AT Madry et al. (2017), TRADES Zhang et al. (2019) and MART Wang et al. (2019) respectively.

**Hyperparameters.** The parameter $\alpha$ in Eq. 3 is determined heuristically and is set to specific values for each of our methods. We study the influence of the $\alpha$ hyperparameter on the natural and robust accuracy for MWPB-AT on CIFAR-10. The study is carried out using WRN-34-10. Our observations are provided in Appendix A. For MWPB-AT, MWPB-TRADES, and MWPB-MART on CIFAR-10, $\alpha$ is set to 0.58, 0.42, and 0.55, respectively. On Tiny Imagenet, we set $\alpha$ to 0.55, 0.4, and 0.7 for MWPB-AT, MWPB-TRADES, and MWPB-MART, respectively. Finally, for MWPB-AT, MWPB-TRADES, and MWPB-MART in other scenarios, we set $\alpha$ to 0.5, 0.15, and 0.6, respectively.

**Baselines.** Our baselines include Standard AT Madry et al. (2017), TRADES Zhang et al. (2019), and MART Wang et al. (2019). Furthermore, we conduct a comparative analysis of our approach against MMA Ding et al. (2019), which also introduces adaptive perturbation radii to enhance adversarial robustness. Lastly, we compare our results to other works that utilize logit-margin for improving adversarial robustness *MAIL* Liu et al. (2021) and *WAT* Zeng et al. (2021). All the hyperparameters of the baselines are the same as in their original papers. Nevertheless, we maintain consistency by using the same learning rate, batch size, and weight decay values as those utilized during the training of our proposed method.

## 5.2 THREAT MODELS

We assess the performance of the proposed method attacks under *White-box* and *Black-box* settings and *Auto attack*.

**White-box attacks.** These attacks have access to model parameters. To evaluate robustness on CIFAR-10 using RN-18 and WRN34-10, we apply the PGD attack with $\epsilon = 8/255$, step size $\kappa$ = $1/255$, $K = 20$; CW (CW loss (Carlini & Wagner, 2017) optimized by PGD-20) attack with $\epsilon = 8/255$, step size $1/255$. On SVHN and Tiny Imagenet, we apply PGD attack with $\epsilon = 8/255$, step size $\kappa = 1/255$, $K = 100$.

**Black-box attacks.** An adversary does not have access to the model parameters under black-box settings. We tested the robust models trained on CIFAR-10 against strong black-box attacks Square (Andriushchenko et al., 2020) with 5,000 queries and SPSA (Uesato et al., 2018) with 100 iterations, perturbation size of 0.001 (for gradient estimation), learning rate = 0.01, and 256 samples for each gradient estimation. All black-box evaluations are made on trained WRN34-10.

**Auto attacks.** Lastly, we evaluated the trained models on *Autoattack* Croce & Hein (2020b), which is a powerful ensemble of attacks consisting of APGD-CE Croce & Hein (2020b), APGD-T Croce & Hein (2020b), FAB-T Croce & Hein (2020a), and Square (a black-box attack) Andriushchenko et al. (2020) attacks.

Table 1: Comparing white-box attack robustness (accuracy %) for RN18 on CIFAR-10. We exclude the standard deviations of three runs since they are insignificant ($< 0.3$) .

| DEFENSE | NATURAL | PGD-20 | CW | AUTOATTACK |
|---|---|---|---|---|
| AT | **84.10** | 52.72 | 51.80 | 47.95 |
| MWPB-AT | 83.78 | **56.25** | **53.02** | **49.96** |
| MART | 80.32 | 55.15 | 49.35 | 47.63 |
| MWPB-MART | **82.23** | **57.10** | **52.57** | **49.53** |
| TRADES | 82.65 | 52.82 | 51.82 | 48.96 |
| MWPB-TRADES | **82.89** | **55.53** | **53.04** | **50.73** |

Table 2: Comparing white-box attack robustness (accuracy %) for WRN34-10 on CIFAR-10. We exclude the standard deviations of three runs since they are insignificant ($< 0.3$)

| DEFENSE | NATURAL | PGD-20 | CW | AUTOATTACK | SQUARE | SPSA |
|---|---|---|---|---|---|---|
| AT | 86.21 | 56.12 | 54.95 | 51.92 | 60.12 | 61.05 |
| MWPB-AT | **86.82** | **59.18** | **57.36** | **54.16** | **61.15** | **63.07** |
| MART | 84.17 | 58.10 | 54.51 | 51.11 | 58.74 | 58.91 |
| MWPB-MART | **85.70** | **60.65** | **56.78** | **53.80** | **60.83** | **62.02** |
| TRADES | 84.70 | 56.30 | 54.51 | 53.06 | 59.16 | 61.15 |
| MWPB-TRADES | **86.09** | **59.10** | **57.04** | **54.38** | **60.77** | **62.19** |

Table 3: Comparing white-box attack robustness (accuracy %) for RN18 on SVHN. We exclude the standard deviations of three runs since they are insignificant ($< 0.3$)

| DEFENSE | NATURAL | PGD-20 | CW | AUTOATTACK |
|---|---|---|---|---|
| AT | **92.67** | 55.67 | 52.92 | 45.94 |
| MWPB-AT | 91.45 | **61.81** | **55.89** | **49.73** |
| MART | **91.59** | 58.78 | 52.79 | 43.60 |
| MWPB-MART | 91.51 | **61.87** | **55.11** | **48.45** |
| TRADES | **90.65** | 57.27 | 53.59 | 46.45 |
| MWPB-TRADES | 90.30 | **60.25** | **55.03** | **50.11** |

Table 4: Comparing white-box attack robustness (accuracy %) for RN18 on Tiny Imagenet. We exclude the standard deviations of three runs since they are insignificant ($< 0.3$)

| DEFENSE | NATURAL | PGD-20 | CW | AUTOATTACK |
|---|---|---|---|---|
| AT | 48.83 | 23.96 | 21.85 | 17.91 |
| MWPB-AT | **51.21** | **25.07** | **23.11** | **19.75** |
| MART | 46.01 | 26.03 | 21.78 | 19.18 |
| MWPB-MART | **47.39** | **27.15** | **22.89** | **20.51** |
| TRADES | 49.11 | 22.82 | 17.79 | 16.82 |
| MWPB-TRADES | **52.12** | **24.60** | **19.85** | **18.15** |

## 5.3 PERFORMANCE EVALUATION

We summarize our results on CIFAR-10 using RN18 and WRN34-10 in Tables 1 and 2 respectively. In addition, we report results on SVHN and Tiny Imagenet using RN18 in Tables 3 and 4.

### 5.3.1 COMPARISON WITH VANILLA BASELINES

We compared our proposed method with vanilla baselines standard AT Madry et al. (2017), TRADES Zhang et al. (2019) and MART Wang et al. (2019). Experimental results demonstrate that the introduction of *MWPB* leads to enhancements in *AT*, *TRADES*, and *MART*. Moreover, our proposed method exhibits improvements in robust accuracy without compromising natural accuracy. These performance gains are consistent across different datasets and baselines. Specifically, when combined with *AT*, *MWPB-AT* showcases notable improvements against adversarial attacks PGD-20 (+3.06), CW (+2.41), and Autoattack (+2.24) on CIFAR-10 when using WRN34-10. Similarly, on datasets SVHN and Tiny Imagenet with RN18, *MWPB-AT* outperforms *AT* against PGD-20, CW, and Autoattack. The performance of *MWPB* remains consistent when integrated with *TRADES* and *MART*. *MWPB-TRADES* exhibits enhancements in both natural accuracy and robustness against attacks PGD, CW, and Autoattack. Similarly, *MWPB-MART* shows considerable improvements, especially against Autoattack, on CIFAR-10 when using WRN-34-10 These improvements extend to datasets SVHN and Tiny Imagenet with RN-18.

Lastly, experimental results also show that our method improves performance on strong black-box attacks Square and SPSA. The results are summarized in Table 2.

Table 5: Comparing white-box and black-box attack robustness (accuracy %) of various margin-based approaches for WRN34-10 on CIFAR-10, and other prominent baselines. We exclude the standard deviations of three runs since they are insignificant ($< 0.3$).

| DEFENSE | NATURAL | PGD-20 | CW | AUTOATTACK | SPSA |
|---|---|---|---|---|---|
| MMA (DING ET AL. (2019)) | 86.29 | 57.12 | **57.59** | 44.52 | 59.87 |
| WAT (ZENG ET AL. (2021)) | 85.13 | 56.63 | 53.97 | 50.01 | 60.75 |
| MAIL (LIU ET AL. (2021)) | 86 .81 | **60.49** | 51.45 | 47.11 | 59.25 |
| **MWPB-AT (OURS)** | **86.85** | 59.18 | 57.36 | **54.16** | **63.07** |
| GAIRAT ZHANG ET AL. (2020) | 85.41 | 60.76 | 45.02 | 42.29 | 52.32 |
| AWP WU ET AL. (2020) | 85.36 | 58.04 | 55.92 | 53.92 | 62.57 |
| **MWPB-AWP** | **87.61** | **61.56** | **58.54** | **56.02** | **64.11** |
| ST-AT LI ET AL. (2023) | 84.91 | 57.52 | 55.11 | 53.54 | 61.34 |

### 5.3.2 COMPARISON WITH OTHER ADAPTIVE RADII AND MARGIN-BASED METHODS

We compare our proposed method to *MMA* Ding et al. (2019), which also aims to improve adversarial training by enabling the adaptive selection of the "correct" perturbation radii. *MMA* minimizes adversarial losses at the "shortest possible perturbation" for individual examples. Experimental results displayed in Table 5 show that the proposed method outperforms *MMA* on natural accuracy (+0.46), PGD-20 (+2.06), Auto attack (+9.64), and SPSA (+3.2), albeit *MMA* slightly performs better on CW attack (-0.33). It's important to highlight that *MMA* employs a bisection search algorithm to determine optimal perturbation radii for adversarial examples, whereas our approach involves a simpler reweighting of the commonly used perturbation radius. This difference in methodology is worth noting, as the bisection search employed by *MMA* can be computationally more expensive. Furthermore, our method utilizes lower mean perturbation radii compared to *MMA* as shown in Table 6 to achieve better robust accuracies. Table 6 shows the average perturbation radii across adversarial examples used for training the best performing epoch.

Existing works Zeng et al. (2021) and Liu et al. (2021) have incorporated the idea of multi-class margin specified in Eq. (2) to improve adversarial robustness. Specifically, these methods utilize multi-class logit margins to reweight adversarial losses, assigning larger weights to losses corresponding to easily misclassified adversarial examples. Our experimental results in Table 5

Table 6: Comparing mean perturbation used for training on CIFAR-10 on best epoch.

| DEFENSE | MEAN RADII |
|---|---|
| MWPB-AT | 0.043 |
| MWPB-TRADES | **0.037** |
| MWPB-MART | 0.039 |
| MMA (DING ET AL. (2019)) | 0.047 |

show that our methods perform better than these methods on stronger attacks CW and Auto attack. Also, these methods have been argued to show signs of gradient obfuscation Fakorede et al. (2023b), given their low performance on strong black-box attacks.

## 6 CONCLUSION

In this paper, we argue that natural examples, from which adversarial examples are generated, exhibit differing levels of inherent vulnerabilities. As a result, we advocate against the use of uniform perturbations in the inner maximization step of the adversarial training framework. Rather, we propose an instance-specific weighting function for determining the perturbation radii when creating individual adversarial examples for adversarial training. The weighting function assesses the susceptibility of each natural example and utilizes this data as a parameter for determining perturbation radii when generating adversarial examples. Experimental results show that our proposed approach consistently enhances the performance of popular adversarial training methods across various datasets and under different attacks. Furthermore, our method demonstrates improvements compared to related adaptive radii and margin-based methods.

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

# A APPENDIX

## A.1 ROBUSTNESS AGAINST PGD-20 UNDER VARIOUS PERTURBATION SIZES AND OTHER ATTACKS

We study the robustness of the proposed method under various $l_\infty$-bounded perturbation sizes. In addition, we evaluate the robustness of the proposed method against $l_2$ autoattacks under the perturbation size of 128/255. In addition, we tested against a strong attack, FAB, which adaptively tracks the decision boundary with the aim of changing the class of a given input. Our experimental results are shown in Table 7.

Table 7: Comparing white-box attack robustness (accuracy %) for WRN-34-10 on CIFAR-10 under various $l_\infty$-bounded perturbation sizes for PGD-20 and Autoattack ($L_2$). We exclude the standard deviations of three runs since they are insignificant ($< 0.3$).

| DEFENSE | PGD-20 $\epsilon = 6/255$ | PGD-20 $\epsilon = 10/255$ | PGD-20 $\epsilon = 12/255$ | AA ($L_2$) $\epsilon = 128/255$ | FAB |
|---|---|---|---|---|---|
| AT | 64.49 | 49.58 | 45.29 | 58.52 | 52.02 |
| MWPB-AT | **67.12** | **52.08** | **48.81** | **61.09** | **54.52** |
| MART | 65.41 | 52.31 | 48.84 | 57.75 | 52.18 |
| MWPB-MART | **67.19** | **55.63** | **51.96** | **60.41** | **54.25** |
| TRADES | 64.19 | 51.92 | 48.23 | 58.05 | 53.59 |
| MWPB-TRADES | **65.82** | **53.58** | **50.11** | **60.03** | **55.17** |

## A.2 IMPACT OF $\alpha$ HYPERPARAMETER.

Here, we study the impact of $\alpha$ in the reweighting function proposed in Eq. 3. The hyperparameter $\alpha$ controls the weight of the reweighting function. Our experiments observed that when $\alpha$ is set low, the natural accuracy is high, but the robust accuracy is quite low and comparable to the standard AT. The robust accuracy on PGD-20 and Auto-attack improves as we increase the value of $\alpha$. At the same time, if $\alpha$ is set too high, the natural accuracy suffers. This trade-off informed our selection of the best value for $\alpha$. We select values that ensure the proper balance between natural and robust accuracy.

Table 8: Studies on *MWPB-AT* showing the impact of the $\alpha$ hyperparameter.

| $\alpha$ | NATURAL | PGD-20 | AA |
|---|---|---|---|
| 0.10 | 88.93 | 56.25 | 51.67 |
| 0.20 | 88.56 | 56.85 | 51.88 |
| 0.30 | 88.18 | 57.80 | 52.58 |
| 0.40 | 87.72 | 58.11 | 52.85 |
| 0.50 | 87.40 | 58.78 | 53.64 |
| **0.58** | **86.82** | **59.18** | **54.16** |
| 0.70 | 85.55 | 59.12 | 54.14 |
| 0.85 | 85.11 | 59.39 | 54.02 |
| 1.2 | 83.75 | 59.69 | 53.50 |

## A.3 DISTRIBUTION OF PERTURBATION RADII

We show in Fig. 2, 3 and 4, the perturbation radii distribution for MWPR-AT, MWPB-MART, and MWPB-TRADES for RN-18 over the 50,000 training samples of CIFAR-10. The perturbation radii distribution is computed on the best-performing epoch in each case. Experimental results show that MWPB-AT has utilized the minimum perturbation radii of 0.018 and the maximum perturbation radii of 0.055. MWPB-MART and MWPB-TRADES have minimum perturbation radii of 0.020 and 0.0225, respectively. The maximum perturbation radii of MWPB-MART and MWPB-TRADES are respectively 0.0554 and 0.0471.

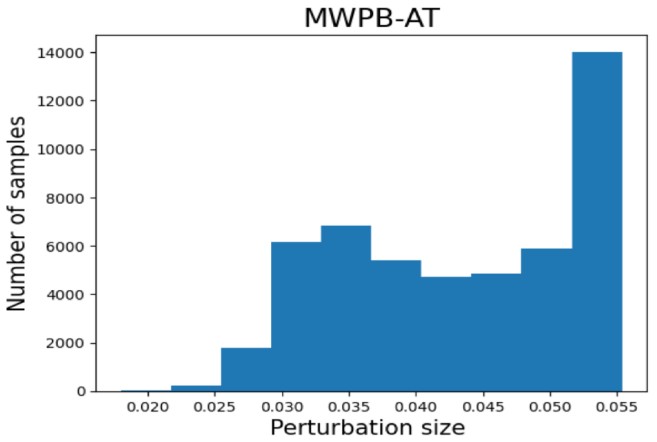

Figure 2: Plot showing the distribution of perturbation radii for MWPB-AT.

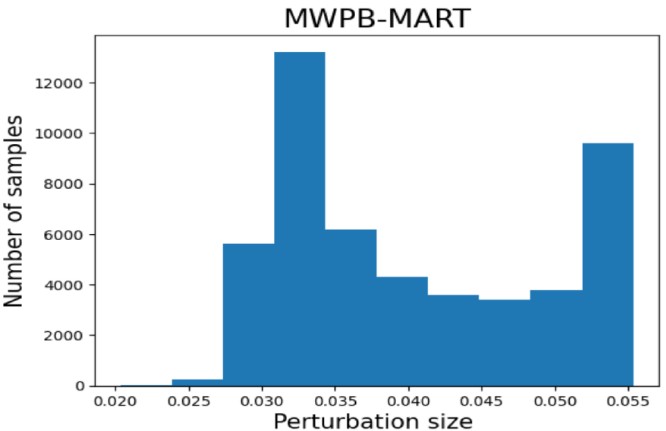

Figure 3: Plot showing the distribution of perturbation radii MWPB-MART.

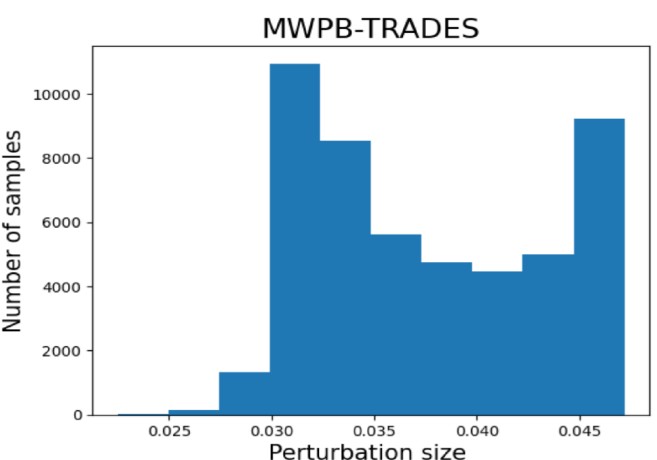

Figure 4: Plot showing the distribution of perturbation radii for MWPB-TRADES.

## B  APPENDIX

The PGD attack utilized for inner maximization is a first-order adversary Madry et al. (2017); Simon-Gabriel et al. (2019).

**Lemma 1 (Pang et al. (2020))** *Given a loss function $\mathcal{L}$ and under the first-order Taylor expansion, the solution to the inner maximization:*

$$\max_{\boldsymbol{x}' \in B_\epsilon(\boldsymbol{x})} L(f_\theta(\boldsymbol{x}'), y)$$

*is $x^* = x + \epsilon \mathbb{U}_p(\nabla_x \mathcal{L}(x))$. Furthermore, $\mathbf{L}(f(x^*), y) = \mathcal{L}(f(x), y) + \epsilon \|\nabla_x \mathcal{L}(x)\|_q$, where $\|.\|_q$ is the dual norm of $\|.\|_p$*

**Lemma 2 (Katharopoulos & Fleuret (2017))** *Let $\mathcal{L}$ be either a negative log-likelihood or square error loss function. Then, $\mathcal{L}(f_\theta(x_1), y_1) > \mathcal{L}(f_\theta(x_2), y_2) \iff \|\nabla_x \mathcal{L}(f_\theta(x_1), y_1)\| > \|\nabla_x \mathcal{L}(f_\theta(x_2), y_2)\|$.*

*Proof of Theorem 1*

**Proof 1** *We anchor the proof of Theorem 1 on Lemma 1 and Lemma 2.*

*Given two input-label pairs $(x_1, y_1)$ and $(x_2, y_2)$, such that $\mathcal{L}(f_\theta(x_1), y) > \mathcal{L}(f_\theta(x_2), y)$. Then according to Lemma 1, the inner maximization of $\mathcal{L}(f_\theta(x_1), y_1)$ and $\mathcal{L}(f_\theta(x_2), y_2)$ under the same perturbation bound $\epsilon$ are given as:*

$$\mathcal{L}(f_\theta(x_1^*), y_1) = \mathcal{L}(f_\theta(x_1), y_1) + \epsilon \|\nabla_{x_1} \mathcal{L} f_\theta(x_1, y_1)\|_q$$

$$\mathcal{L}(f_\theta(x_2^*), y_2) = \mathcal{L}(f_\theta(x_2), y_2) + \epsilon \|\nabla_{x_2} \mathcal{L} f_\theta(x_2, y_2)\|_q$$

*It can seen that the increment in loss resulting from the inner maximization depend on $\epsilon \|\nabla_{x_1} \mathcal{L}(f_\theta(x_1), y_2)\|_q$ and $\epsilon \|\nabla_{x_2} \mathcal{L}(f_\theta(x_2), y_2)\|_q$. Since $\epsilon$ is constant then the loss increment depend on norm of the gradient. Also according to Lemma 2, $\|\nabla_x L(f_\theta(x_1), y_1)\|_q > \|\nabla_x L(f_\theta(x_2), y_2)\|_q$. Therefore under a fixed perturbation radius $\epsilon$,*

1. *$\mathcal{L}(f_\theta(x_1^*), y_1) > \mathcal{L}(f_\theta(x_2^*), y_2)$*

2. *$\mathcal{L}(f_\theta(x_1^*), y_1) - \mathcal{L}(f_\theta(x_1), y_1) > \mathcal{L}(f_\theta(x_2^*), y_2) - \mathcal{L}(f_\theta(x_2), y_2)$*

*Proof of Theorem 2*

**Proof 2** *Consider an input-label pair $(x, y)$ and a loss function $\mathcal{L}$ with the inner maximization*

$$\max_{\boldsymbol{x}' \in B_\epsilon(\boldsymbol{x})} L(f_\theta(\boldsymbol{x}'), y)$$

*and the solution*

$$L(f(x^*), y) = L(f(x), y) + \epsilon \|\nabla_x L(f_\theta(\boldsymbol{x}), y)\|_q$$

*. Then, given a perturbation radius $\epsilon_g > \epsilon$, we have:*

$$L(f(x^*), y) = L(f(x), y) + \epsilon_g \|\nabla_x L(f_\theta(\boldsymbol{x}), y)\|_q$$

*. Since $L(f(x), y)$ and $\|\nabla_x L(f_\theta(\boldsymbol{x}'), y)\|_q$ remain fixed, and $\epsilon_g > \epsilon$, then $L(f(x^*), y)$ computed within $\epsilon_g$ is greater than $L(f(x^*), y)$ within $\epsilon$.*

## C APPENDIX

Here we present the proposed algorithm for MWPB-AT

---

**Algorithm 1** MWPB-AT Algorithm.

---
**Input:** a neural network model with the parameters $\theta$, step sizes $\kappa_i$ and $\kappa$, and a training dataset $\mathcal{D}$ of size n.

**Output:** a robust model with parameters $\theta^*$

1: **set** $\epsilon = 8/255$
2: **for** $epoch = 1$ to num_epochs **do**
3:    **for** $batch = 1$ to num_batchs **do**
4:       sample a mini-batch $\{(x_i, y_i)\}_{i=1}^{M}$ from $\mathcal{D}$;             $\triangleright$ mini-batch of size $M$.
5:       **for** $i = 1$ to M **do**
6:          $d_{margin}(\mathbf{x}_i, y_i; \theta) = f_\theta(\mathbf{x}_i)_{y_i} - \max_{j, j \neq y_i} f_\theta(\mathbf{x}_i)_j$
7:          $\epsilon_i = exp(\alpha \cdot d_{margin}(\mathbf{x}_i, y_i; \theta)) * \epsilon$
8:          $\kappa_i = \epsilon_i/4$
9:          $\mathbf{x}_i^{'} \leftarrow \mathbf{x}_i + 0.001 \cdot \mathcal{N}(0, 1)$; $\triangleright \mathcal{N}(0, I)$ is a Gaussian distribution with zero mean and identity variance.
10:          **for** $k = 1$ to $K$ **do**
11:             **if** $epoch \leq 80$ **then**
12:                $\mathbf{x}_i' \leftarrow \prod_{B_{\epsilon/2}(\mathbf{x}_i)} (x_i + \kappa/2 \cdot sign(\nabla_{\mathbf{x}_i'} L(f_\theta(\mathbf{x}_i'), y_i))$;    $\triangleright \prod$ is a projection operator.
13:             **else**
14:                $\mathbf{x}_i' \leftarrow \prod_{B_{\epsilon_i}(\mathbf{x}_i)} (x_i + \kappa_i \cdot sign(\nabla_{\mathbf{x}_i'} L(f_\theta(\mathbf{x}_i'), y_i))$
15:             **end if**
16:          **end for**
17:       **end for**
18:       $\theta \leftarrow \theta - \kappa \nabla_\theta \sum_{i=1}^{M} L(f_\theta(\mathbf{x}_i'), y_i)$
19:    **end for**
20: **end for**

---

