# OpenReview forum: "IMPROVING ADVERSARIAL TRAINING WITH MARGIN- WEIGHTED PERTURBATION BUDGET"
_ICLR.cc/2024/Conference — ICLR 2024 Conference Withdrawn Submission_

### Official Review · Reviewer_Sshb · 2023-10-25

**Soundness:** 2 fair
**Presentation:** 3 good
**Contribution:** 2 fair
**Rating:** 5
**Confidence:** 4

**Summary:**

The paper focused on the sample importance in adversarial training (AT). The authors argue that adversarial examples (AEs) generated from inherently robust natural samples might not induce a sufficiently high loss to achieve optimal robustness, and thus crafting AEs with a fixed perturbation budget may limit the effectiveness of AT. They used the proximity of the natural samples to the class boundaries to distinguish important samples and proposed a reweighting function for assigning different perturbation budgets to individual AEs during AT. Experiments show the superiority over baselines.

**Strengths:**

1. The whole logic of this paper is easy to follow. The objective function and the realization of the proposed method are clear.
2. The authors proposed a realized measurement to compute the distance to the decision boundary, which is simple and intuitive.
3. Experimental results show that, compared to the baseline approaches, this method can achieve a notable improvement in robustness and accuracy simultaneously on CIFAR-10 and Tiny Imagenet.

**Weaknesses:**

1. The paper has limited novelty at the philosophy level. [1] and [2] pointed out that from a geometric perspective, a natural data point closer/farther from the class boundary is less/more robust, and the weight of the adversarial data should be assigned accordingly. The proposed MWPB shares the same philosophy. The only difference is that this paper focuses on weighted perturbation budget while [1] and [2] focus on weighted robust loss. Besides, [3] and [4] have already proposed instance-wise AT methods of using weighted perturbation budgets 4 years ago. For now, the idea of using instance-wise weighted perturbation budgets in AT is no longer novel. At least, the authors should justify the necessity of using MWPB and explain why MWPB is better than the above-mentioned methods.
2. The experiments are insufficient. Since the main idea of this paper is to justify the necessity of using adaptive $\epsilon_i$ in AT, the authors should at least try different $\epsilon$ in the setting of $\ell_\infty$-norm (e.g., 6/255, 10/255, 12/255) to see if MWPB can still work well. The authors should at least examine the range of $\epsilon$ within which MWPB can outperform other methods. Furthermore, the authors should discuss why this paper only considers $\ell_\infty$-norm, instead of $\ell_2$-norm. I would suggest the authors to try $\epsilon$ in the setting of $\ell_2$-norm (e.g., 128/255) as well.
3. Besides, the authors should test the robustness of the proposed method under adaptive attacks [5]. Specifically, adaptive attacks assume the attackers know all the details of MWPB, including the proposed reweighting function. Since MWPB decreases the perturbation radii for vulnerable natural data, I wonder if the generated AEs from these vulnerable natural data are sufficiently robust. Therefore, I would suggest proposing an adaptive attack that assigns a larger perturbation radii to vulnerable natural data than robust ones during the generation of AEs at the test time. Then, see if MWPB can successfully defend against such attack and still outperform other methods.
4. The paper has several typos and formatting inconsistencies. For example, in Section 4.4, it should be ‘Large perturbation size … find sharper minima’ instead of ‘shaper minima’, and ‘Equation 4’ should be ‘Eq. 4’ to ensure the consistency. In Table 4, ‘27.15’ should be in **bold** instead of *italic*. Besides, Table 5 and Table 7 exceed the prescribed margins. I recommend a thorough review and correction of these errors to ensure the professionalism of the paper.

[1] Jingfeng Zhang, Jianing Zhu, Gang Niu, Bo Han, Masashi Sugiyama, and Mohan Kankanhalli. Geometry-aware instance-reweighted adversarial training. In *International Conference on Learning Representations*, 2020.

[2] Huimin Zeng, Chen Zhu, Tom Goldstein, and Furong Huang. Are adversarial examples created equal? a learnable weighted minimax risk for robustness under non-uniform attacks. In *Proceedings of the AAAI Conference on Artificial Intelligence*, volume 35, pp. 10815–10823, 2021.

[3] Gavin Weiguang Ding, Yash Sharma, Kry Yik Chau Lui, and Ruitong Huang. Mma training: Direct input space margin maximization through adversarial training. In *International Conference on Learning Representations*, 2019.

[4] Yogesh Balaji, Tom Goldstein, and Judy Hoffman. Instance adaptive adversarial training: Improved accuracy tradeoffs in neural nets. *arXiv preprint arXiv:1910.08051*, 2019.

[5] Anish Athalye, Nicholas Carlini, and David A. Wagner. Obfuscated gradients give a false sense of security: Circumventing defenses to adversarial examples. In *International Conference on Machine Learning*, 2018.

**Questions:**

1. Theorem 1 points out inherently robust data contribute less to the inner maximization compared to inherently vulnerable data. Theorem 2 shows that increasing $\epsilon$ will increase the inner maximization. Therefore, it is reasonable to give a larger $\epsilon$ to those robust data. But is it necessary to decrease $\epsilon$ to inherently vulnerable data? According to theorem 2, decreasing $\epsilon$ will decrease the inner maximization. Could you provide a theoretical justification for why it is beneficial to decrease the perturbation radii on vulnerable natural examples?
2. In Section 5.1 Hyperparameters, I noticed that the hyperparameter $\alpha$ is different for every method (e.g., MWPB-AT, MWPB-TRADES and MWPB-MART) and dataset (e.g., CIFAR-10 and Tiny Imagenet). As a result, I assume the performance of MWPB heavily relies on the choice of $\alpha$. If $\alpha$ has to be manually chosen for every method and dataset, I believe it will bring some additional cost. Then it is probably not correct to say ‘our proposed method reweights perturbation radii at no additional cost’ in Section 2.3. Could you please further clarify how efficient it is to find an optimal hyperparameter $\alpha$?

Regarding the notable improvement in robustness and accuracy, I will consider increasing my score if the authors can address all my suggestions and questions well.

---

> ### Author Response · Authors · 2023-11-21
>
> We thank the reviewer for the valuable feedback.
>
>
>
> **"The paper has limited novelty at the philosophy level. [1] and [2] pointed out that from a geometric perspective..."**
>
>
> The novelty of this work is in showing that better robustness may be obtained by perturbing by adaptively assigning perturbation radii to individual samples used for AT. To the best of our knowledge, no existing work has clearly demonstrated this observation.
>  [1] and [2] assign larger weights to the losses of adversarial data close to the decision boundary. In contrast, we propose assigning larger perturbation budgets for crafting adversarial examples originating from natural data farther from the decision boundary. While these works and ours both utilize geometric information, the core philosophy of those ideas differs from ours. Also, [1] and [2], do not yield genuine adversarial robustness as they perform poorly on strong attacks [6, 7]. To the best of our knowledge, our work is the first reweighting approach to yield genuine robustness to stronger attacks like Autoattacks.
>
>
>
>
>
> **"Besides, [3] and [4] have already proposed instance-wise AT methods of using weighted perturbation budgets 4 years ago. At least, the authors should justify the necessity of using MWPB and explain why MWPB is better than the above-mentioned methods."**
>
> It should be noted that [3] and [4] did not exactly use a “weighted” perturbation budget, although these works proposed training with adversarial examples crafted using unequal perturbation budgets.  Besides, none of these methods show that robustness may be improved by training with adaptive perturbation radii. We have highlighted the differences in Sec. 2.3.
>
>
>
> **"The authors should at least try different $\epsilon$ in the setting of ℓ∞-norm (e.g., 6/255, 10/255, 12/255) to see if MWPB can still work well."**
>
>
> We have revised the initial submission to include this. Kindly check Appendix A.1.
>
>
>
>
> **"The authors should discuss why this paper only considers ℓ∞-norm, instead of  ℓ2-norm."**
>
>
> We followed the trend in adversarial robustness literature which prioritizes ℓ∞-norm. However, based on your suggestion, we have added experiments on ℓ2-norm Autoattack. Kindly check Appendix A.1 of the revised paper.
>
>
>
> **"Besides, the authors should test the robustness of the proposed method under adaptive attacks [5]. Specifically, adaptive attacks assume the attackers know all the details of MWPB, including the proposed reweighting function."**
>
>
>
> We could not implement the new adaptive attack the reviewer described within this time frame. However, we evaluated our defense against a strong adaptive boundary attack called FAB, and we have added the result in Appendix A.1. FAB attack is also resistant to gradient masking. Notwithstanding, our hunch is that MWPB would perform better under the adaptive attack you described, partly because vulnerable samples are usually in the minority. In addition, vulnerable samples usually contribute less to robust accuracy in traditional AT methods as may be seen in [7] and [8].
>
>
>
> **"Several typos and formatting inconsistencies. "**
>
> We have corrected the typos and revised the whole paper.
>
>
>
> **"Could you provide a theoretical justification for why it is beneficial to decrease the perturbation radii on vulnerable natural examples?"**
>
>  The assumption here is that intrinsically natural vulnerable samples are inherently located in areas of high losses. Hence, relatively smaller perturbation radii will be sufficient to obtain a useful inner maximization loss. Since we are reweighting perturbation radii based on the individual vulnerability of natural samples, it makes sense that the radii of vulnerable samples are smaller $\epsilon$. Furthermore, the natural accuracy benefits from smaller perturbation radii assigned to vulnerable examples.
>
>
>
>
> **"If $\alpha$ has to be manually chosen for every method and dataset, I believe it will bring some additional cost. Then it is probably not correct to say ‘our proposed method reweights perturbation radii at no additional cost’ in Sec 2.3."**
>
> By “additional cost,” we mean that the training time for MWPB-AT, MWPB-TRADES, and MWPB-MART does not increase over AT, TRADES, and MART, respectively. Indeed, we manually selected $\alpha$ (See. Table 8, Appendix A.2). It is common to tune for additional parameters in ML literature, e.g., AWP, TRADES, MART, etc. We recommend the ideal values of $\alpha$ for obtaining the best performance and for easy reproduction of results.
>
>
>
> References
>
> 6. Dong, C., Liu, L. and Shang, J., 2021. Data quality matters for adversarial training: An empirical study.
>
> 7. Fakorede, O., Nirala, A.K., Atsague, M. and Tian, J., 2023. Vulnerability-Aware Instance Reweighting For Adversarial Training. TMLR.
>
> 8. Xu, H., Liu, X., Li, Y., Jain, A. and Tang, J., 2021, July. To be robust or to be fair: Towards fairness in adversarial training.  ICLR

---

> ### Comment · Reviewer_Sshb · 2023-11-21
>
> I thank the authors for their detailed responses. Most of my concerns have been addressed and I have raised my score from 3 to 5. I still have some further questions:
>
> 1. After the revision, I am still not convinced by the authors that the novelty of this paper is sufficient. In my opinion, **inspiration is more important than implementation.** Technically, I can see the difference between this work and previous works (e.g., [1] [2] [3] [4]). However, I can hardly see a clear difference in terms of how this idea could inspire more researchers to come up with a follow-up idea. I hope the authors could further clarify this point.
>
> 2. I have read other rebuttals and I am curious about the statement made by the authors: *'Ideally, larger perturbation budgets lower the natural accuracy. However, this was alleviated by starting the training with a smaller perturbation budget ($\epsilon$=4/255) and step size of 1/255, until the 80th epoch.'* My question is: are all the baseline methods following the same experiment setup, i.e., $\epsilon$=4/255 and step size of 1/255 until the 80th epoch? If not, I do not think the comparison is fair enough. I hope the authors could further clarify this point.

---

> ### Author Response · Authors · 2023-11-22
>
> *We thank the reviewer for the insightful comments, and we appreciate the reviewer for increasing our score. *
>
> **“After the revision, I am still not convinced by the authors that the novelty of this paper is sufficient. In my opinion, inspiration is more important than implementation. Technically, I can see the difference between this work and previous works (e.g., [1] [2] [3] [4]). However, I can hardly see a clear difference in terms of how this idea could inspire more researchers to come up with a follow-up idea. I hope the authors could further clarify this point.”**
> To the best of our knowledge, no prior work has convincingly demonstrated that unequal perturbation radii can improve robustness. [1,2,3,4] do not achieve genuine robustness [6, 7]. Reweighting losses, as done in [1] and [2] is not very beneficial [6, 7].
>
> More importantly, our idea is not tied to a specific implementation or algorithm. One of the arguments in our work is that unequal perturbation can benefit any AT method. This line of reasoning is beneficial to the community.
>
>
>
> **“I have read other rebuttals and I am curious about the statement made by the authors: 'Ideally, larger perturbation budgets lower the natural accuracy. However, this was alleviated by starting the training with a smaller perturbation budget ($\epsilon$=4/255) and step size of 1/255, until the 80th epoch.' My question is: are all the baseline methods following the same experiment setup, i.e., $\epsilon$=4/255 and step size of 1/255 until the 80th epoch? If not, I do not think the comparison is fair enough. I hope the authors could further clarify this point.”**
>
> Training with $ \epsilon $ =4/255 and step size of 1/255 until the 80th epoch is part of  the uniqueness of our method. This approach was inspired by the theoretical analysis in [9], which shows that AT under larger perturbation budgets hinders models from escaping sub-optimal initial regions. We use this approach to ease the introduction of larger perturbation radii in the later epochs.
>
> We indeed tested this setup in the baselines that use uniform perturbation and found the setup does not benefit these baselines. Other baselines that change perturbation radii or change weights have their own experiment setup.
>
>
> **Reference**
> [9]. Liu, C., Salzmann, M., Lin, T., Tomioka, R. and Süsstrunk, S., 2020. On the loss landscape of adversarial training: Identifying challenges and how to overcome them. Advances in Neural Information Processing Systems, 33, pp.21476-21487.

---

### Official Review · Reviewer_hkHv · 2023-10-28

**Soundness:** 3 good
**Presentation:** 2 fair
**Contribution:** 2 fair
**Rating:** 3
**Confidence:** 4

**Summary:**

This paper employs a margin-weighted perturbation budget to generate adversarial examples for adversarial training, where the instance-wise weighting is estimated based on the degree of intrinsic vulnerabilities of natural examples.

**Strengths:**

1. Propose an adversarial training method with instance-wise perturbation budget;
2. Achieve robustness improvements across multiple baselines.

**Weaknesses:**

1. Writing needs improvement. There are many confused and hard-to-follow long sentences in the manuscript.
2. Novelty is limited: the motivation is not novel, as the unequal treatment of samples has been extensively discussed in previous research. Methodologically, it relies on a typical combination of reweighting and adaptive perturbation radii, lacking novelty.
3. It is strongly recommended that the authors provide empirical experimental data regarding Theorem 1. For instance, in the case of PGD-AT on the CIFAR-10 dataset, what proportion of samples align with Theorem 1?
4. Empirically, in the later stages of adversarial training, nearly all samples have a d_margin greater than 0. This implies that the perturbation budget for all adversarial samples exceeds 8/255. Given this scenario, achieving higher natural accuracy is quite perplexing. Therefore, it is hoped that the authors can offer detailed insights into the evolution of the perturbation budget during training.
5. Insufficient experiment: it is suggested that the authors conduct experimental comparisons with a broader range of baselines, such as AWP, rather than solely focusing on standard AT methods.

**Questions:**

Please refer to the questions in the Weaknesses.

---

> ### Author Response · Authors · 2023-11-21
>
> We thank the reviewer for the valuable feedback. We have addressed the requested changes in the revision. For easy reference, we have used blue color to mark major changes in the text of the revised paper.
>
> **"Writing needs improvement"**
>
> We have gone through the whole manuscript all over again to improve the presentation and  address the typos.
>
>
>
>  **"Novelty is limited: the motivation is not novel, as the unequal treatment of samples has been extensively discussed in previous research.**
>
>  The novelty of this work is in showing that better robustness may be obtained by perturbing by adaptively assigning perturbation radii to individual samples used for AT. To the best of our knowledge, no existing work has clearly demonstrated this observation.
>  Furthermore, we agree with the reviewer that unequal treatment of samples has been demonstrated in previous research. However, it should be noted that the unequal treatment of samples alone is not an end goal, but why are individual samples unequally treated, and in what context? The context here is different from what is obtainable in existing works. For instance, most existing reweighting approaches focus mainly on reweighting losses by typically assigning larger weights to difficult samples . Here, we assign smaller perturbation radii to difficult samples and we also argue that existing AT methods such as TRADES and MART could benefit from assigning larger perturbation radii to inherently robust samples. Existing instance-wise reweighting methods fail to yield genuine improved robustness as they perform poorly against stronger attacks such as auto-attacks and sometimes show signs of gradient obfuscation [1, 2]. To the best of our knowledge, our work is the first instance-wise reweighting method to yield genuine improvement against auto-attacks, and it does so cheaply.
>
>
>
> **"Empirically, in the later stages of adversarial training, nearly all samples have a d_margin greater than 0. This implies that the perturbation budget for all adversarial samples exceeds 8/255. Given this scenario, achieving higher natural accuracy is quite perplexing. Therefore, it is hoped that the authors can offer detailed insights into the evolution of the perturbation budget during training."**
>
> Ideally, larger perturbation budgets lower the natural accuracy. However, this was alleviated by starting the training with a smaller perturbation budget ($\epsilon$ =4/255) and step size of 1/255, until the 80th epoch. This approach was inspired by the theoretical analysis in [3], which shows that AT under larger perturbation budgets hinders models from escaping sub-optimal initial regions. Our experimental results show how our training approach mitigates low natural accuracy in the proposed MWPB.  For example, the experiments below were obtained for WRN-34-10 on CIFAR-10.
>
>
>
> | Method              	| Natural Accuracy (%) 	| PGD-20 (%)| AA(%)
> |-----------------------|-----------------------|------------------|--------------------|
> | AT (Madry et al, 2018)   		|   86.21   	|                  56.21          		|     51.92            |
> | MWPB-AT (no burn-in period)  		|  84.39   	|     58.71  		|        53.87             |
> | MWPB-AT (with burn-in)  		|  86.85  	|     59.18  		|        54.16             |
>
> Note: We have added the perturbation radii distribution for all training samples of CIFAR-10 on ResNet-18 in Appendix A.3.
>
>
>
>  **"Insufficient experiment: it is suggested that the authors conduct experimental comparisons with a broader range of baselines, such as AWP, rather than solely focusing on standard AT methods."**
>
>
> The main goal of our work is to show that the proposed approach can improve existing AT methods. However, based on your recommendation, we have included new baselines in Table 6. We added the following baselines: AWP,  GAIRAT, and ST-AT. We have also added more experiments under various settings in Table 7.
>
> **References**
>
> 1. Dong, C., Liu, L. and Shang, J., 2021. Data quality matters for adversarial training: An empirical study. arXiv preprint arXiv:2102.07437.
>
> 2. Fakorede, O., Nirala, A.K., Atsague, M. and Tian, J., 2023. Vulnerability-Aware Instance Reweighting For Adversarial Training. TMLR.
>
> 3. Liu, C., Salzmann, M., Lin, T., Tomioka, R. and Süsstrunk, S., 2020. On the loss landscape of adversarial training: Identifying challenges and how to overcome them. Advances in Neural Information Processing Systems, 33, pp.21476-21487.

---

### Official Review · Reviewer_vNfc · 2023-10-30

**Soundness:** 3 good
**Presentation:** 4 excellent
**Contribution:** 3 good
**Rating:** 6
**Confidence:** 4

**Summary:**

This paper introduces a new method to improve the adversarial robustness of DNNs. The authors notice the difference in robustness and their effects on adversarial training between different training samples. Due to the vulnerability of some samples, the attack cannot find the optimal solution for some robust samples. Therefore, they propose to assign larger perturbation budgets to robust samples to enhance the utility of adversarial training. Experiments have shown that the proposed method can improve the robustness.

**Strengths:**

- The authors novelly propose to assign different perturbation budgets to different training samples according to the vulnerability of samples, and they propose an efficient way to represent the vulnerability of samples.
- Experiments show that the DNNs trained with the proposed MWPB exhibit high robustness under various attacks.
- The paper is well-written and easy to follow.

**Weaknesses:**

- How will the proposed method affect the natural accuracy in theory? If the perturbation budgets on robust samples are too large, training on such adversarial examples may hurt the discrimination power and generalization power of the DNN. For example, the natural accuracies are all reduced in Table 3. The authors are suggested also to discuss the theoretical effect of the proposed method on the natural accuracy of the DNN.
- Is there a threshold for the radius in Equation (3)? If not, when an input is too far away from the decision boundary (misclassified or correctly classified), the radius will be extremely small or large, which may affect the stability of training. Besides, the magnitude of outputs $f_\theta(x)$ may be different in different models, so the weight $\alpha$ needs to be carefully adjusted for each model.
- The hyperparameters in experiments are determined heuristically for different methods on different models. This setting hurts the reliability of the obtained results. I wonder whether the proposed method is stable to different hyperparameters.
- In experiments, the proposed method is only used after 80 epochs, at which point the model may have learned a certain optimization direction. What about the performance when using the MWPB from the beginning of the training?

**Questions:**

What is the distribution of the perturbation radius computed by equation (3)? How different is it from the traditional uniform perturbations? In other words, how many samples are assigned small radius and how many samples are assigned large radius?

---

> ### Author Response · Authors · 2023-11-21
>
> We thank the reviewer for the valuable feedback. We have addressed the requested changes in the revision. For easy reference, we have used blue color to mark major changes in the text of the revised paper.
>
> **"How will the proposed method affect the natural accuracy in theory? If the perturbation budgets on robust samples are too large, training on such adversarial examples may hurt the discrimination power and generalization power of the DNN. For example, the natural accuracies are all reduced in Table 3. The authors are suggested also to discuss the theoretical effect of the proposed method on the natural accuracy of the DNN."**
>
> Usually, larger perturbation budgets result in lower natural accuracy. We attempt to  mitigate this  by starting the training with a smaller perturbation budget ($\epsilon$ =4/255) until the 80th epoch. Our experimental results in Tables 1, 2 and 4 show that our approach achieves superior performance and generalization over the baselines.  However, in Table 3, there is a marginal decrease in the natural accuracy. We attribute this to the peculiarity of the SVHN datasets. SVHN is not very favorable to AT.
>
> **The hyperparameters in experiments are determined heuristically for different methods on different models. This setting hurts the reliability of the obtained results. I wonder whether the proposed method is stable to different hyperparameters.**
>
> The proposed MWPB uses one hyperparameter, which is $\alpha$. Since we are trying to assign larger weight perturbation radii to intrinsically robust examples, it is important that $\alpha$ > 0.0. Setting $\alpha$ between 0.0 and 1.0 yields stable results. However, we recommend the values stated in the paper for reproducing the results reported in the paper. Kindly see Table 8 in  Appendix A.2.
>
> **Is there a threshold for the radius in Equation (3)? If not, when an input is too far away from the decision boundary (misclassified or correctly classified), the radius will be extremely small or large, which may affect the stability of training. Besides, the magnitude of outputs  $f_{\theta}(x)$ may be different in different models, so the weight $\alpha$ needs to be carefully adjusted for each model.**
>
> There is no threshold for the radius in Equation (3). The determination of the appropriate radii for individual  samples depend significantly on the logit margin computed from the model’s outputs on natural samples. Based on our observations, the perturbation radii used by MWPB-AT (uses the largest perturbation) for training ResNet-18 on CIFAR-10 ranges between 0.018 and 0.055. The mean perturbation radius for each implementation of MWPB is provided in Table 6. We have also added plots showing the distribution of the perturbation radii in Appendix A.3.
>
> We do not expect  the magnitude of the logit  margins to be significantly different for different models. Based   on our experiments reported in Tables 1  and 2 for ResNet-18 and Wideresnet-34-10 respectively, the best performances were recorded on the same values of $\alpha$ for MWPB-AT, MWPB-TRADES and MWPB-MART respectively.
>
> **In experiments, the proposed method is only used after 80 epochs, at which point the model may have learned a certain optimization direction. What about the performance when using the MWPB from the beginning of the training?**
>
> Ideally, larger perturbation budgets lower the natural accuracy. However, this was alleviated by starting the training with a smaller perturbation budget ($\epsilon$ =4/255) and step size of 1/255, until the 80th epoch. This approach was inspired by the theoretical analysis in [1], which shows that AT under larger perturbation budgets hinders models from escaping sub-optimal initial regions. Our experimental results show how our training approach mitigates low natural accuracy in the proposed MWPB.  For example, the experiments below were obtained for WRN-34-10 on CIFAR-10.
>
>
>
>
>
> | Method              	| Natural Accuracy (%) 	| PGD-20 (%)| AA(%)
> |-----------------------|-----------------------|------------------|--------------------|
> | AT (Madry et al, 2018)   		|   86.21   	|                  56.21          		|     51.92            |
> | MWPB-AT (no burn-in period)  		|  84.39   	|     58.71  		|        53.87             |
> | MWPB-AT (with burn-in)  		|  86.85  	|     59.18  		|        54.16             |

---

### Official Review · Reviewer_jD4X · 2023-10-30

**Soundness:** 2 fair
**Presentation:** 3 good
**Contribution:** 2 fair
**Rating:** 5
**Confidence:** 4

**Summary:**

This paper proposes a reweighting function for assigning perturbation bounds to adversarial examples used in adversarial training, and leads to a new approach named Margin-Weighted Perturbation Budget (MWPB). MWPB assigns perturbation radii to each adversarial sample based on its vulnerability of the corresponding natural example. Empirical evaluations are done on the CIFAR-10, SVHN, and TinyImageNet datasets, and applying ResNet-18/WRN-34-10 model architectures. The results show that combining MWPB can improve robustness under PGD-20, CW, and AutoAttack.

**Strengths:**

The strengths of this paper include:
- Clear writing with some intuitive explanations such as Figure 1.
- The proposed MWPB is straightforward and easy to implement, and it is naturally compatible with different AT frameworks.
- The proposed methods are evaluated under some standard attacking methods such as PGD and AutoAttack, and empirically MWPB can improve robustness compared to baselines.

**Weaknesses:**

The weaknesses of this paper include:
- The construction of the reweighing function in Eq (3) seems heuristic. There should be ablation studies on the effects of different reweighing alternatives, and justify why the construction in Eq (3) is superior.
- The considered baselines (AT, MART, TRADES) are not up-to-date, where the top-rank methods listed on RobustBench [1] can already achieve >70% robust accuracy under AutoAttack on CIFAR-10 ($\ell_{\infty}=8/255$). The authors are encouraged to validate the compatibility of MWPB with more advanced methods listed on RobustBench.
- The values of hyperparameters in Section 5.1 seem to be deliberately selected (e.g., $\alpha=0.58,0.42,0.55$). There should be ablation studies on the effects of different hyperparameters on the performance of MWPB (i.e., is MWPB sensitive to different values of hyperparameters?).

[1] https://robustbench.github.io/

**Questions:**

As stated in the Weaknesses sections, the baselines (AT, MART, TRADES) are not up-to-date (proposed no later than 2019). So I would like to know if MWPB can be compatible with more advanced methods listed on RobustBench. Besides, there should be ablation studies on the effects of different reweighing alternatives and sensitivity w.r.t. hyperparameters.

---

> ### Author Response · Authors · 2023-11-21
>
> We thank the reviewer for the valuable feedback. We have addressed the requested changes in the revision. For easy reference, we have used blue color to mark major changes in the text of the revised paper.
>
> **The construction of the reweighting function in Eq (3) seems heuristic. There should be ablation studies on the effects of different reweighing alternatives, and justify why the construction in Eq (3) is superior.**
>
> We have added ablation studies in the revised version. Kindly check Table 8 in Appendix A.2.
>
> **The considered baselines (AT, MART, TRADES) are not up-to-date, where the top-rank methods listed on RobustBench [1] can already achieve >70% robust accuracy under AutoAttack on CIFAR-10 ( ℓ∞=8/255). The authors are encouraged to validate the compatibility of MWPB with more advanced methods listed on RobustBench.**
>
> One of the goals of our work is to show that this approach can improve prominent existing works such as AT, TRADES and MART.  In addition, we have compared our results with more baselines in Table 5.
> It is important to note that the robustness reported for the top-rank methods in  Robustbench [1] are significantly dependent on a combination of the network architecture (WRN-74-10), the amount of training data (50 million), the activation function(swish), batch size, and the number of training epochs ( typically > 1000) . Due to computational resources constraints, we could not evaluate on WRN-74-10 and train on 50 million data.  Nevertheless, we combined MWPB-TRADES with [2]. Due to constraints in computational resources, we can only execute 400 epochs instead of the recommended 2,400 epochs, even with  20 million data  generated by EDM. Also, the evaluation was done  using WRN-28-10.
> Our experimental results are as follows:
>
>
>
> | Method              	| NAT (%) 	|AA(%)|
> |-----------------------|-----------------------|------------------|
> | [2]  		|  91.47   	|     63.84  		|
> | MWPB-TRADES  		|    90.31	|     63.78
> | MWPB-AT   		|  91.12   	|     64.11
>
>
> It should be noted the batch size used in [2] is 2048, whereas we used 512. It is reported in [2] that the batch size plays a significant role in the robust accuracy. In fact, the best accuracy reported in [2] are trained on a batch size of 2048. To achieve the best robustness of over 70%, [2] trained WRN-74-10 for 2400 epochs.
>
> **The values of hyperparameters in Section 5.1 seem to be deliberately selected (e.g., $\alpha$=0.58,0.42,0.55). There should be ablation studies on the effects of different hyperparameters on the performance of MWPB (i.e., is MWPB sensitive to different values of hyperparameters?).**
> Kindly check Appendix A.2 to see the sensitivity of the hyperparameter $\alpha$.
>
> **As stated in the Weaknesses sections, the baselines (AT, MART, TRADES) are not up-to-date (proposed no later than 2019). So I would like to know if MWPB can be compatible with more advanced methods listed on RobustBench. Besides, there should be ablation studies on the effects of different reweighing alternatives and sensitivity w.r.t. hyperparameters.**
> We have added ablation studies on Appendix A.2. We have also added newer baselines in Table 5.
>
> **References**
>
> [1] Robustbench
>
> [2]. Zekai Wang, Tianyu Pang, Chao Du, Min Lin, Weiwei Liu, and Shuicheng Yan. Better diffusion models further improve adversarial training. ICML, 2023.

---

> > ### Comment · Reviewer_jD4X · 2023-11-21
> >
> > I thank the authors for their responses and new experiments, which are helpful.
> >
> > However, I find that the reported comparison with [2] seems misleading. The authors mentioned that [2] uses *20M EDM data, 2048 bs, and 2400 epochs*, however, under theses setups, the result of [2] on WRN-28-10 is 92.44 \% NAT and 67.31 \% AA as listed on RobustBench. These results are far higher than the results reported by the authors.
> >
> > Besides, under the setups of *1M EDM data, 512 bs, and 400 epochs*, [2, Table 2] reports 91.12 \% NAT and 63.35 \% AA on WRN-28-10. It seems that combining MWPB-TRADES with [2] does not lead to significant improvements (90.31 \% NAT and 63.78 \% AA).
> >
> > Could the authors further clarify these points?

---

> ### Author Response · Authors · 2023-11-21
>
> *We thank the reviewer for the timely response. *
>
>
> **"I find that the reported comparison with [2] seems misleading. The authors mentioned that [2] uses 20M EDM data, 2048 bs, and 2400 epochs, however, under theses setups, the result of [2] on WRN-28-10 is 92.44 % NAT and 67.31 % AA as listed on RobustBench".**
>
> Truely, [2] obtained 67.31% AA under the settings you stated. However, due to the computational cost involved, we did not train our model using these settings. We trained our model using the batch size 512 and for 400 epochs. The reported comparison with [2] is from **Table 5** in **[2]** using **20M EDM data, 2048 bs, and 400 epochs which reported 63.84% on AA.**
>
>
>
>
> **"Besides, under the setups of 1M EDM data, 512 bs, and 400 epochs, [2, Table 2] reports 91.12 % NAT and 63.35 % AA on WRN-28-10. It seems that combining MWPB-TRADES with [2] does not lead to significant improvements (90.31 % NAT and 63.78 % AA)"**.
>
>
> [2] reported 91.47% NAT 63.84% AA under 20M EDM data, 2048 bs, and 400 epochs, while combining MWPB-TRADES with [2] obtains 90.31 % NAT and 63.78 % AA under 20M EDM data, 512 bs, and 400 epochs. By the information in [2], batch size plays an important role in robustness accuracy, which makes us believe that MWPB-TRADES could improve over 63.84% when a batch size of 2048 is used.   **Kindly note that we did not train on the setup of 1M EDM data **.

---

### Comment · Area_Chair_oFnG · 2023-11-21
**[Time Sensitive, ICLR24] Please read the authors' responses and try to discuss the remaining concerns with the authors**

Dear Reviewers,

The authors have provided detailed responses to your comments.

Could you have a look and try to discuss the remaining concerns with the authors? The reviewer-author discussion will end in two days.

We do hope the reviewer-author discussion can be effective in clarifying unnecessary misunderstandings between reviewers and the authors.

Best regards,

Your AC